# Seeing Things: A Community Science Investigation into Motion Illusion Susceptibility in Domestic Cats (*Felis silvestris catus*) and Dogs (*Canis lupus familiaris*)

**DOI:** 10.3390/ani12243562

**Published:** 2022-12-16

**Authors:** Gabriella E. Smith, Philippe A. Chouinard, Isabel Lin, Ka Tak Tsoi, Christian Agrillo, Sarah-Elizabeth Byosiere

**Affiliations:** 1Thinking Dog Center, Department of Psychology, Hunter College, The City University of New York, New York, NY 10065, USA; 2Messerli Research Institute, University of Veterinary Medicine Vienna, 1210 Vienna, Austria; 3School of Psychology and Public Health, La Trobe University, Bundoora 3086, Australia; 4Bronx High School of Science, New York, NY 10468, USA; 5Department of General Psychology, University of Padova, 35122 Padova, Italy; 6Padua Neuroscience Center, University of Padova, 35131 Padova, Italy

**Keywords:** illusion, dog, animal cognition, perception, cat, Rotating Snakes, motion

## Abstract

**Simple Summary:**

The study of visual illusion susceptibility offers a fascinating lens into the evolution of perception. Utilizing a community science paradigm, this study investigated pet dogs’ and cats’ susceptibility to the Rotating Snakes motion illusion. The results reveal that both species did not spend significantly more time at the illusion than at either of the controls, failing to indicate susceptibility to the illusion. These findings offer valuable information for the field of non-human animal geometric illusion research, both in terms of comparative perception and methodological practices.

**Abstract:**

Illusions—visual fields that distort perception—can inform the understanding of visual perception and its evolution. An example of one such illusion, the Rotating Snakes illusion, causes the perception of motion in a series of static concentric circles. The current study investigated pet dogs’ and cats’ perception of the Rotating Snakes illusion in a community science paradigm. The results reveal that neither species spent significantly more time at the illusion than at either of the controls, failing to indicate susceptibility to the illusion. Specific behavioral data at each stimulus reveal that the most common behaviors of both species were Inactive and Stationary, while Locomotion and Pawing were the least common, supporting the finding that susceptibility may not be present. This study is the first to examine susceptibility to the Rotating Snakes illusion in dogs, as well as to directly compare the phenomenon between dogs and cats. We suggest future studies might consider exploring alternative methods in testing susceptibility to motion illusions in non-human animals.

## 1. Introduction

Visual illusions—visual fields that trick the brain into perceiving environmental features that are absent—are powerful tools to investigate perception. Studies investigating visual illusion susceptibility in humans have examined the perception of illusory features including, but not limited to, shape (e.g., impossible trident, [1]), size (e.g., Müller-Lyer, [2]), coloration (e.g., checker shadow illusion, [3]), and motion (e.g., peripheral drift illusion, [4]) (see Ref. [5]). Susceptibility to visual illusions has also been examined in non-human animals, with the first studies dating back to the 1920s [6]. Since then, research has found surprising results, such as some species perceiving illusions in similar or opposite ways to humans [7], and that some experimental methods may influence apparent perception [8].

Investigation into the possible perception of visual illusions in non-human animals typically relies on two methods [8]. One method involves operant training, in which a subject is trained to indicate where the target illusion is observed. For example, Bravo et al. (1998) [9] trained domestic cats (*Felis silvestris catus*) to select among various manipulations of the Kanizsa square where they perceived the subjective square illusion to be. An alternative experimental method involves spontaneous choice, in which an individual exhibits attraction to an illusion due to its biological relevance (most often food). A study by Smith et al. (2020) [10] re-examined domestic cats’ perception of the Kanizsa square by presenting cats with the illusion, a square, and a control, relying on cats’ attraction to enclosed spaces to determine the illusory perception. The authors found that the subjects stood/sat in the illusion as much as in the square and more than in the control, confirming that cats demonstrate a human-like susceptibility to the illusion. Studies such as these prove useful in revealing potentially analogous, if not identical, modes of perception between species, and, thus, inform theories regarding the evolution of perception. Furthermore, replicated investigation utilizing varying methodologies offers valuable insight into the influence of procedure in purported findings.

A third method of studying illusory susceptibility is through physiological responsiveness, a method seen best in studies of motion illusion perception. For example, with regards to the Rotating Snakes illusion (a series of static concentric circles varying in color), human participants exhibit ocular saccades in response to peripheral drift caused by the illusion [11] (occurring in color and in grayscale versions [12]). While there lacks consensus as to how the perception of the Rotating Snakes illusion truly occurs, one theory posits that it is caused by luminance differences between the higher and lower contrast spots of the illusion [11]. Human-like perception of motion in the Rotating Snakes illusion has been documented in diverse vertebrate species, such as non-human primates [13] and fish [14], the latter of which also appears to exhibit ocular saccades in response to the Rotating Snakes illusion. Such susceptibility to this illusion has been theorized to be due to shared neurocognitive architecture across phyla [13,15].

The effects of the Rotating Snakes illusion on behavioral responsiveness have also been studied in cats. In 2014, an investigation by Bååth et al. [16] utilized a self-report paradigm in which owners presented the illusion and a control stimulus to pet cats and reported preferential looking and hunting behaviors toward the illusion. Despite promising results, variables such as limited data and the lack of a controlled environment suggest that more research may be required to support the reported susceptibility. A later study by Regaiolli et al. (2019) [17] examined the perception of the Rotating Snakes illusion in a captive, zoo-housed, group of lions (*Panthera leo*) and found that two out of the three lions spent more time at the illusion than the control stimuli, indicating susceptibility. The authors also observed the lions exhibit natural behaviors, such as predatory biting and dragging of all three stimuli, as well as toward other environmental stimuli, indicating the effect of the stimuli on stimulating hunting behaviors. Thus, the results of these studies suggest cats make an excellent model to examine the perception of the Rotating Snakes illusion due to the illusion’s perceived motion potentially triggering cats’ attraction to stimuli that resemble the movements of small prey [18].

Another popular companion animal species, dogs, are also descended from carnivorous predators, wolves [19], and may also exhibit hunting behaviors stimulated by the perceived motion of the Rotating Snakes illusion. While many visual illusions have been examined in dogs (*Canis lupus familiaris*) (see Ref. [20] for a review), susceptibility to the Rotating Snakes illusion has not yet been examined in any member of the Canidae family. This study, thus, aims to examine susceptibility to the Rotating Snakes illusion in both cats and dogs, examining both the time spent at the illusion compared to the controls and the specific behaviors expressed toward the stimuli.

The pets in this community science study were presented with three stimuli (illusion and two controls) and were observed in their home environment using a spontaneous behavior paradigm in the first comparative approach to Rotating Snakes susceptibility in companion animals. Data from both dogs and cats offer not only an interesting species comparison into motion illusion susceptibility to aid the understanding of the evolution of perception, but also methodological efficacy of the spontaneous choice paradigm. While little work has explicitly compared the visual systems of dogs and cats, what is currently known is that, compared to cats, dogs may have a greater minimum threshold of light for vision as well as potentially being less efficient at reflecting light off their tapetums, both likely due to their differing circadian visual systems [21]. Regarding visual acuity, while cats are suspected to have fewer nerve fibers in their optic nerves than dogs do, the two species are suspected to have similar visual acuity [21]. Due to this and the theorized conservation of the neurocognitive architecture necessary to perceive the illusion among vertebrates [13,15], we hypothesize that both cats and dogs are susceptible to the illusion. We, thus, predict the subjects will spend more time at the illusion than at the controls, as well as potentially express hunting behaviors (e.g., pawing) toward the perceived motion.

## 2. Materials and Methods

### 2.1. Justification and Stimuli

This four-month study took place from May to September 2021 and was adapted from Regaiolli et al.’s (2019) [17] methodology into a community science paradigm due to its efficacy in examining non-human animal behavior and cognition (e.g., [10,22,23]). This study was approved by CUNY Hunter College Institutional Animal Care and Use Committee (SEB—Seeing Things 10/23 Do You See What I See? A Companion Animal Community Science Project). IRB review was not required by CUNY Hunter College Human Research Protection Program (HRPP) as no identifiable information from the pet owner was collected for research purposes.

The three stimuli presented were: Rotating Snakes; a control stimulus with the same pattern as the illusion that does not elicit perceived motion (hereafter called Plain Snakes); and a control stimulus consisting of a simple pattern of overlapping circles that also does not elicit perceived motion (hereafter called Big Circles) (Figure 1). The Rotating Snakes illusion consists of alternating black, blue, white, and yellow blobs (black, dark grey, white, light grey for the achromatic version). Plain Snakes (Control 1) does not evoke any motion perception, even though the overall configuration is identical to that of the Rotating Snakes pattern. Since the order of the colored segments in the Plain Snakes stimulus is reversed between adjacent blobs (relative to the Rotating Snakes), the local motion signal is nulled, acting as a useful tool to assess whether animals are attracted by the apparent movement of the Rotating Snakes or simply a complex visual pattern. Big Circles (Control 2) is another control pattern differing from the first two visual stimuli since it consists of overlapping circles, but the overall stimulus is extremely less complex than the other stimuli and does not invoke the perception of motion.

### 2.2. Participants

Participation was recruited through flyers advertised on social media and other forms of community outreach. Overall, 122 pet owners signed up their pets to participate (72 cats and 50 dogs), and, of these, 39 cats and 36 dogs completed and were included in the study. Cat and dog participants were not limited by age, breed, size, or other demographics, on the condition they were not visually impaired. Participants were not compensated but received a congratulatory certificate following completion of the study.

### 2.3. Demographic Questions

Owners were instructed to fill out a Qualtrics survey with information including the animal’s sex, age, breed, and spayed/neutered status. Other intake questions included when the owners acquired their pet, from where, and whether the animal had any known visual disabilities.

### 2.4. Design

Owners were emailed detailed instructions to perform the experiment, specifically how to present stimuli to their pets and film all behaviors in the following ten minutes. Videos recorded by owners were uploaded to a Dropbox link provided by the researchers. The three stimuli (25.2 cm × 18.75 cm each) were made available for download in color (see Appendix A) or in black and white (see Appendix A) depending on the owners’ preferred printer ink colors. Participants were randomly assigned one of six different orders of stimuli presentation to control for any side or order preferences: (1) Rotating Snakes/Plain Snakes/Big Circles; (2) Plain Snakes/Rotating Snakes/Big Circles; (3) Plain Snakes/Big Circles/Rotating Snakes; (4) Big Circles/Rotating Snakes/Plain Snakes; (5) Big Circles/Plain Snakes/Rotating Snakes; (6) Rotating Snakes/Big Circles/Plain Snakes.

### 2.5. Procedure

The study was performed in individual participants’ homes. The three stimuli were taped 61 cm (2 feet) apart from one another on the floor, ideally in the middle of a room (Figure 2). Owners were instructed to wear dark sunglasses and not interact with their pet when releasing them into the room and while video recording them for ten minutes.

### 2.6. Analysis

Videos were coded in their entirety using BORIS coding software (Version 8) [24] by two coders (I.L., and K.T.; 99% intercoder reliability via intraclass correlation coefficient). Coded state behaviors were adapted based on pre-existing research [17,25]: Locomotion: The subject has an obvious active orientation toward a stimulus while continuously moving in a forward direction and has at least one limb within one body length from the stimulus.Stationary: The subject has an obvious active orientation toward and has at least one limb within one body length from the stimulus. The subject is either sitting, standing, or lying down for at least two seconds. If the subject does not move for one minute, coded as inactive (below).Sniffing: The subject is approaching a stimulus at around the width of approximately two human fingers or less using its nose.Pawing: One of the subject’s front limbs is raised and lowered in a quick succession, either over or contacting the stimulus (e.g., Figure 3).Inactive: The subject has a stationary position without any obvious active orientation toward any physical or social stimuli for at least four seconds (e.g., head down; lying down with eyes closed; resting; falling asleep).

Parametric analysis deemed the data to not be normally distributed; thus, non-parametric alternative statistics were conducted. Due to this, Friedman tests were used instead of one-way ANOVAs with repeated measures to examine the effects of the variables (e.g., location; stimulus; ink color) on the time spent during the experiment. Mann–Whitney U tests examined species comparisons, and corrected *p*-values (pcorr) are reported after Bonferroni corrections. Corrected *p*-values (pcorr) were calculated by multiplying the resulting *p*-value by 3. Descriptive statistics were performed to examine behaviors expressed by both species at the different stimuli.

## 3. Results

### 3.1. Time Spent during Experiment

See Figure 4 for graphical representations.

#### 3.1.1. Dogs

Compared to the time spent at the stimuli, dogs spent more time away from the stimuli (*p* = 0.002), but not out of view (*p* = 0.862). Between the time spent away from the stimuli and out of view, dogs spent more time away from the stimuli (*p* < 0.001).

#### 3.1.2. Cats

Compared to the time spent at the stimuli, cats spent more time away (*p* = 0.024), but not out of view (*p* > 0.999). Between the time spent away from the stimuli and out of view, cats spent more time away from the stimuli (*p* = 0.009).

#### 3.1.3. Species Comparison

Between dogs and cats, a significant difference was found in the time spent at the stimuli (U = 955.500, N_Cat_ = 39, N_Dog_ = 36, pcorr = 0.021), with dogs spending more time at the stimuli. However, no significant differences were found between the species in the time spent away (U = 574.500, N_Cat_ = 39, N_Dog_ = 36, pcorr = 0.534), nor out of view (U = 871.000, N_Cat_ = 39, N_Dog_ = 36, pcorr = 0.195).

### 3.2. Time Spent at Stimuli

See Figure 5 for graphical representations. See Appendix A for a qualitative report of the stimuli at which each individual spent the most time (note that this does not consider small differences in time spent). Note that, in Appendix A, the order of orientation options A–F (B and W) and G–L (color) corresponds with orientations 1–6, respectively, described in Section 2.

#### 3.2.1. Dogs

Thirty-one out of thirty-six dog subjects spent time at the stimuli and, thus, contributed to the data analysis. With Bonferroni corrections, no significant differences were found in the time spent between: Big Circles and Plain Snakes (*p* = 0.073) (despite a significant, uncorrected trend toward more time spent at Big Circles, *p* = 0.024); Big Circles and Rotating Snakes (*p* = 0.101) (despite a significant, uncorrected trend toward more time spent at Big Circles, *p* = 0.034); nor Plain Snakes and Rotating Snakes (*p* > 0.999).

#### 3.2.2. Cats

Thirty-eight out of thirty-nine cat subjects spent time at the stimuli and, thus, contributed to the data analysis. With Bonferroni corrections, no significant differences were found in the time spent between: Big Circles and Plain Snakes (*p* = 0.547); Big Circles and Rotating Snakes (*p* > 0.999); nor Plain Snakes and Rotating Snakes (*p* > 0.999).

#### 3.2.3. Species Comparison

Between dogs and cats, no significant differences were found in the time spent at: Big Circles (U = 505.500, N_Cat_ = 38, N_Dog_ = 31, pcorr = 0.939); Plain Snakes (U = 682.500, N_Cat_ = 38, N_Dog_ = 31, pcorr = 0.747); nor Rotating Snakes (U = 678.500, N_Cat_ = 38, N_Dog_ = 31, pcorr = 0.813).

### 3.3. Behaviors at Stimuli

The species appeared consistent in the frequency of behaviors at the stimuli (Figure 6, Table 1). In both dogs and cats, the most common behavior was Inactive, occurring 83.2% of the time in dogs, and 50.28% of the time in cats. The least common behavior in both species was Pawing, occurring 0.40% of the time in dogs and 2.6% of the time in cats.

### 3.4. Effect of Ink Color

The split of species is comparable between the two ink options—black and white ink (B and W) (13 cats and 14 dogs); color ink (26 cats and 22 dogs)—with color ink being the most common overall (Table 2).

#### 3.4.1. On Time Spent during Experiment

In cats, there were no significant effects of the ink type on the time spent: at the stimuli (U = 165.000, N = 39, pcorr > 0.999); away from the stimuli (U = 163.00, N = 39, pcorr > 0.999); nor out of view (U = 142.000, N = 39, pcorr > 0.999). In dogs, there were also no significant effects of the ink type on the time spent: at the stimuli (U = 166.000, N = 36, pcorr > 0.999); away the from stimuli (U = 171.000, N = 36, pcorr > 0.999); nor out of view (U = 137.500, N = 36, pcorr > 0.999).

#### 3.4.2. On time Spent at Stimuli

Again, only those subjects that spent time at the stimuli were considered for calculations relating to the time spent at the stimuli. In cats, there were no significant effects of the ink type on the time spent: at Big Circles (U = 211.000, N = 38, pcorr = 0.414); at Plain Snakes (U = 122.000, N = 38, pcorr = 0.645); nor at Rotating Snakes (U = 137.00, N = 38, pcorr > 0.999). In dogs, there were also no significant effects of the ink type on the time spent: at Big Circles (U = 101.000, N = 31, pcorr > 0.999); at Plain Snakes (U = 121.000, N = 31, pcorr > 0.999); nor at Rotating Snakes (U = 112.000, N = 31, pcorr > 0.999). See Table 3 for a descriptive report evaluating the mean proportion of the time spent at each stimulus for each species by the ink type. 

## 4. Discussion

This study examined susceptibility to the Rotating Snakes motion illusion in domestic dogs and cats. We found that neither species spent more time at the illusion than either of the controls, failing to reveal a susceptibility in dogs and replicate previous positive findings in cat studies [16,17]. While it is possible that susceptibility was not observed, such contradictory results could be methodological in nature. Despite the spontaneous choice paradigm’s effectiveness in revealing illusory susceptibility via subjects’ attraction to perceived motion without the (potentially confounding) use of food, it is also known for its flaws [8,26]. Specifically, subjects may orient more to the illusion due to other factors (e.g., context of food reward) rather than perceived motion. In this study, the opposite occurred: Neither species oriented to the illusion more than the other stimuli, but dogs spent relatively more time at the Big Circles (before Bonferroni corrections) than cats did. Such results can be explained by using similar reasoning: Consisting of several overlapping, solid blocks of color, the Big Circles’ appearance is distinctly different from that of both the Rotating Snakes and Plain Snakes, and, thus, may have attracted slightly more interest. Thus, careful consideration of the context and stimuli appearance is encouraged in study paradigms such as these.

Further contradictory results of this study may also point to other methodological confounds. Specifically, it is possible that the subjects’ view of the stimuli from above was not conducive to perceiving the illusion (e.g., [27]). Furthermore, due to the relatively high area ratio between the stimuli and the rest of the room, it is not surprising the species spent more time away from the stimuli than with them. If performed again, presentation of the stimuli on a vertical plane and control for the experimental “arena” areas may prove more conducive to subject perception and, thus, behavioral expressions of susceptibility. 

Despite the benefits of this community science paradigm, such as a large sample size and the effectiveness of the home setting at stimulating natural behavior, it also saw drawbacks. The biggest limitation of this study was the lack of environmental controls, such as consistency in time of day, environmental brightness, floor type, and placement of the stimuli. Furthermore, since the data are not normally distributed, the data needed to be transformed into ranks, limiting analysis. Additionally, along these lines, was the arbitrariness of the coded data, since behaviors such as attention, time spent nearby, and looking can prove ambiguous. Lastly, large datasets often discount individual data. For example, pawing did occur at the illusion in both dogs (0.22%) and cats (2.37%), albeit infrequently, occurrences of which can be lost when comparing large groups. Thus, both group- and individual-level data comparisons may prove informative in future research.

## 5. Conclusions

Overall, this study failed to support our hypotheses of motion perception in the Rotating Snakes illusion as assessed in both pet dogs and cats. Such results may be due to a lack of susceptibility or be rooted in the drawbacks of the spontaneous choice paradigm and/or physical presentation of the stimuli. Although this study is not conclusive, our data do not encourage the idea that dogs and cats perceive the illusory motion elicited by the Rotating Snakes pattern. However, it is also possible that both species are susceptible to illusory motion, but such an effect is not strong enough to generate any behavioral outcome in both or either species and/or overall preference in favor of the Rotating Snakes stimulus. Future studies using different experimental paradigms (e.g., operant conditioning procedures, such as those used with monkeys [13] and fish [14]) are needed to clarify the current findings. In the absence of such investigations, we remain cautious to claim that these companion animal species are not susceptible to the Rotating Snakes illusion.

## Figures and Tables

**Figure 1 animals-12-03562-f001:**
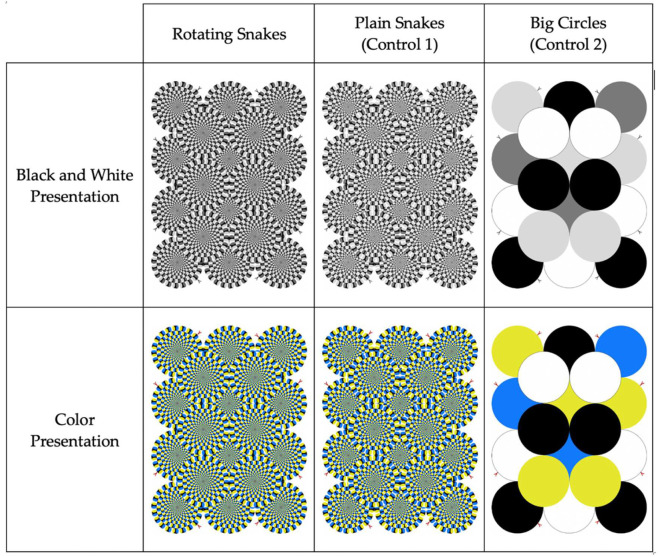
Study stimuli presented to participants, represented both in black and white, and color presentations.

**Figure 2 animals-12-03562-f002:**
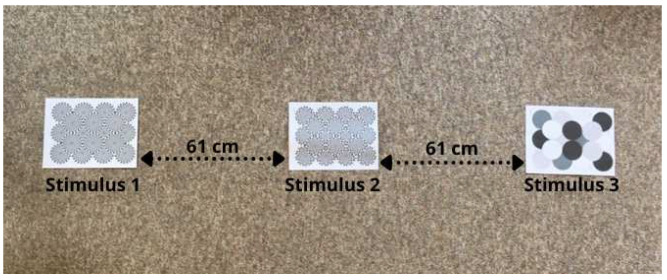
Stimuli presentation.

**Figure 3 animals-12-03562-f003:**
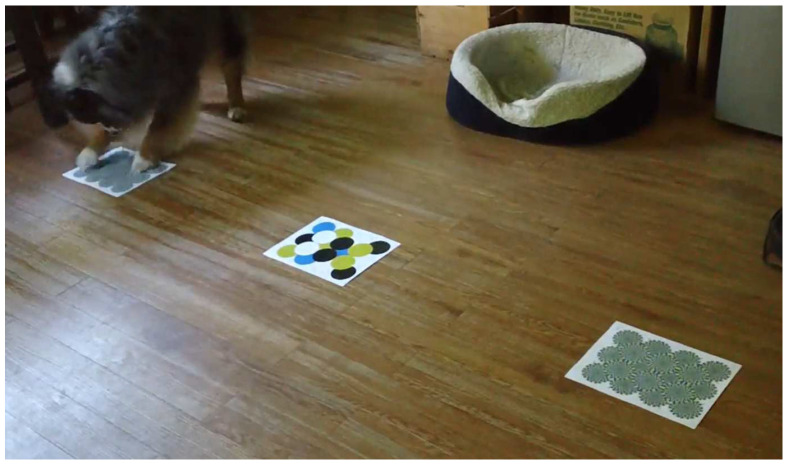
Participant pawing at illusion.

**Figure 4 animals-12-03562-f004:**
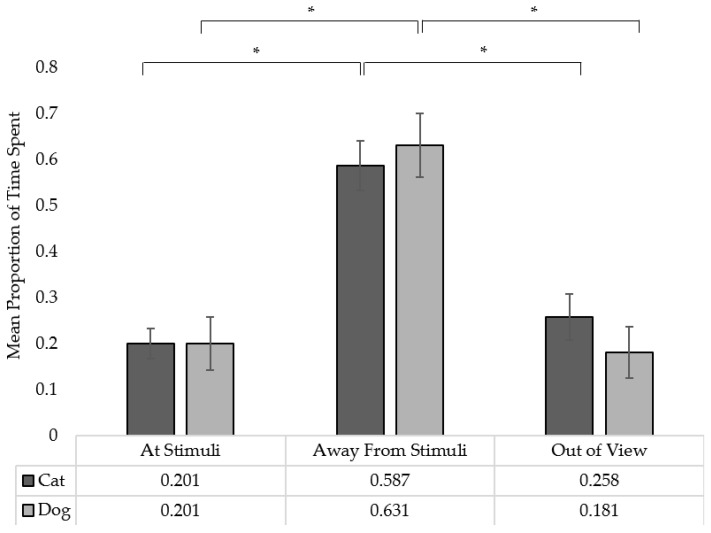
Mean proportion of time spent at the stimuli, away from the stimuli, or out of view relative to the total length of footage. Error bars denote standard errors. * Denotes a statistically significant difference (*p* < 0.5).

**Figure 5 animals-12-03562-f005:**
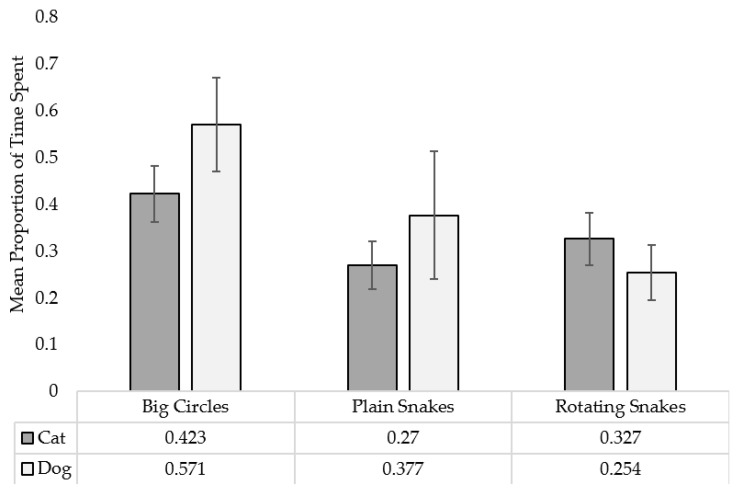
Mean proportion of time spent at each stimulus, the two controls (Big Circles and Plain Snakes) and the illusion (Rotating Snakes) by each species. Error bars denote standard errors.

**Figure 6 animals-12-03562-f006:**
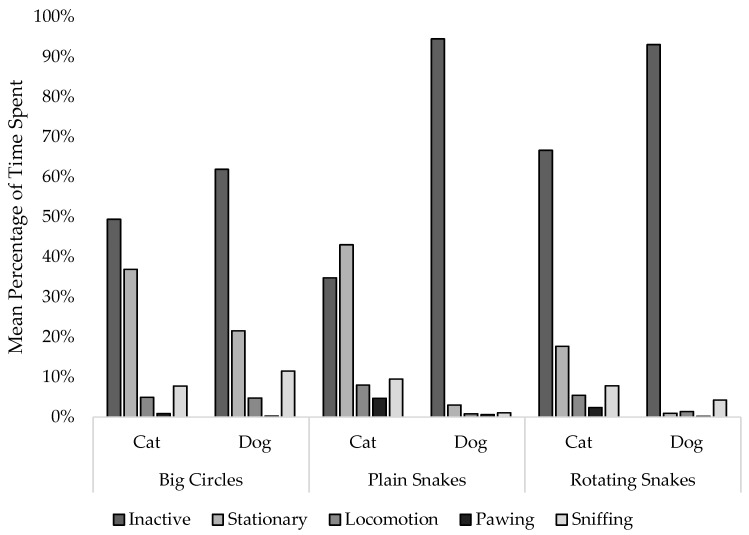
Mean percentage of time spent expressing specific behaviors at each stimulus by species.

**Table 1 animals-12-03562-t001:** Percentage of time spent expressing specific behaviors at stimuli by dogs and cats.

	Cats			Dogs		
Behavior	Big Circles	Plain Snakes	Rotating Snakes	Big Circles	Plain Snakes	Rotating Snakes
Inactive	49.42%	34.78%	66.63%	61.93%	94.47%	93.09%
Locomotion	4.98%	8.04%	5.47%	4.71%	0.82%	1.41%
Pawing	0.88%	4.65%	2.37%	0.29%	0.60%	0.22%
Sniffing	7.80%	9.51%	7.84%	11.46%	1.09%	4.28%
Stationary	36.92%	43.01%	17.69%	21.61%	3.02%	1.00%

**Table 2 animals-12-03562-t002:** Descriptive report of proportion of time spent out of total time per ink group per species.

Species	Location	Ink	N	Mean	SD	SE
Dogs (N = 36)	At Stimuli	B and W	14	0.231	0.400	0.107
		Color	22	0.184	0.311	0.066
	Away from Stimuli	B and W	14	0.665	0.431	0.115
		Color	22	0.610	0.414	0.088
	Out of View	B and W	14	0.097	0.238	0.064
		Color	22	0.235	0.383	0.082
Cats (N = 39)	At Stimuli	B and W	13	0.194	0.183	0.051
		Color	26	0.204	0.212	0.042
	Away from Stimuli	B and W	13	0.585	0.328	0.091
		Color	26	0.588	0.343	0.067
	Out of View	B and W	13	0.254	0.359	0.100
		Color	26	0.260	0.290	0.057

**Table 3 animals-12-03562-t003:** Descriptive report of proportion of time spent per ink group.

Species	Stimulus	Ink	N	Mean	SD	SE
Dogs (N = 31)	Big Circles	B and W	12	0.518	0.520	0.150
		Color	19	0.604	0.597	0.137
	Plain Snakes	B and W	12	0.279	0.376	0.109
		Color	19	0.439	0.931	0.214
	Rotating Snakes	B and W	12	0.283	0.380	0.110
		Color	19	0.236	0.303	0.070
Cats (N = 38)	Big Circles	B and W	13	0.540	0.439	0.122
		Color	25	0.362	0.327	0.065
	Plain Snakes	B and W	13	0.191	0.286	0.079
		Color	25	0.312	0.328	0.066
	Rotating Snakes	B and W	13	0.269	0.337	0.094
		Color	25	0.357	0.352	0.070

## Data Availability

Data can be accessed in the Appendix A.

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
