# Peer review of "Seeing Things: A Community Science Investigation into Motion Illusion Susceptibility in Domestic Cats (Felis silvestris catus) and Dogs (Canis lupus familiaris)"

_animals, 2022, doi:10.3390/ani12243562_

Round 1

Reviewer 1 Report

I commend the authors on using an interesting manipulation to study apparent movement illusion susceptibility in both cats and dogs. A surprising small number of studies directly compare cat and dog cognition, so this is a nice addition to the literature. However, the authors could go a step further to emphasize expected differences between domestic cats and dogs based on differences in the process of domestication for each species and differences in diet. Cats are obligate carnivores whereas dogs are not and domestic pets may have their hunting instincts suppressed through captivity and training so these might be important aspects to address to justify the comparison.  It would be helpful to know more about differences in the visual systems of cats and dogs as well.

The authors should be clear that differential behavior can provide evidence of discrimination but a null result does not indicate a failure to discriminate – it simply shows a lack of preference in this case, which is not the same thing.

I really like the control for complexity. I assume it is not possible to have a less complex pattern that does evoke the motion illusion but it might be helpful to state this explicitly.

The authors should stipulate that the patterns were placed on the floor rather than upright on a wall and how they were affixed (if so).

Why did the authors decide to have all three stimuli presented simultaneously rather than conduct three trials with only one novel stimulus present on each trial? Now the amount of time spent at each is dependent on the time spent on the others and the cats or dogs may just choose one to attend to and ignore the others. Had the stimuli been presented independently, the authors could have conducted an OMNIBUS ANOVA with species and stimulus type and color as factors allowing them to interact. Are Friedman tests appropriate for dependent data? The authors don’t state explicitly if they took the amount of time at each stimulus out of the total time spent with stimuli in general (rather than the total time of the session). Again, they need to be more detailed and explicit in their description of what was done.

Ignoring the less complex stimulus, it looks like there could be an interaction between species and preference for behaviors directed at rotating vs. plain snakes, which is interesting, even if not statistically significant. Perhaps with a different statistical approach such an effect could be examined?

What is the presentation order if all were presented at once? Do the authors mean the placement from left to right? The methods need to be described more explicitly.

What were the coding intervals? A lot of needed detail is lacking if someone wanted to replicate the study.

The authors should also note the lack of standardization of the floor surface on which the stimuli were placed. Ideally, they should be placed on a standard black or white background. I agree with them that presenting the stimuli vertically might be important. The authors should indicate how the stimuli are typically presented in previous studies.

I commend the authors for embarking on citizen science to collect these data.

Reviewer 2 Report

The methodology should be supplemented with a description of the statistical activities carried out, which were scattered by the authors and described in the Results chapter, preceding the result tables (181-189, 207-212, 242-245).

Presentation of results through charts will make it more attractive to present data, especially I see the application of action in the case of percentage data (table 3).

In the chapter Discussion lacks sufficient literature, the authors discuss with only felines (269) without reference to canids. The next item of literature included in the discussion (273) refers to , and the last to reptiles (284). In total, the academic polemic conducted by the authors of the chapter the discussion refers only to four items of literature, in none of which is related to dogs. I consider such consideration of the results obtained in the study to be absolutely insufficient, thus obliging the authors to expand and supplement the chapter with adequate manuscripts and references to the relevant animal species.  The discussion of the research results is far from expected, and references to the obtained results are visible only in percentage terms (297). A description of the results obtained against the background of existing research is necessary. Only supplementing the discussion will allow for reliable inference.
